# Description of Chemical Synthesis, Nuclear Magnetic Resonance Characterization and Biological Activity of Estrane-Based Inhibitors/Activators of Steroidogenesis

**DOI:** 10.3390/molecules28083499

**Published:** 2023-04-15

**Authors:** Donald Poirier

**Affiliations:** 1Laboratory of Medicinal Chemistry, Endocrinology and Nephrology Unit, CHU de Québec Research Center-Université Laval, Québec, QC G1V 4G2, Canada; donald.poirier@crchudequebec.ulaval.ca; 2Department of Molecular Medicine, Faculty of Medicine, Université Laval, Québec, QC G1V 0A6, Canada

**Keywords:** steroid, chemical synthesis, nuclear magnetic resonance, inhibitor, activator, enzyme, 17beta-hydroxysteroid dehydrogenase, steroid sulfatase

## Abstract

Steroid hormones play a crucial role in several aspects of human life, and steroidogenesis is the process by which hormones are produced from cholesterol using several enzymes that work in concert to obtain the appropriate levels of each hormone at the right time. Unfortunately, many diseases, such as cancer, endometriosis, and osteoporosis as examples, are caused by an increase in the production of certain hormones. For these diseases, the use of an inhibitor to block the activity of an enzyme and, in doing so, the production of a key hormone is a proven therapeutic strategy whose development continues. This account-type article focuses on seven inhibitors (compounds **1**–**7**) and an activator (compound **8**) of six enzymes involved in steroidogenesis, namely steroid sulfatase, aldo-keto reductase 1C3, types 1, 2, 3, and 12 of the 17β-hydroxysteroid dehydrogenases. For these steroid derivatives, three topics will be addressed: (1) Their chemical synthesis from the same starting material, estrone, (2) their structural characterization using nuclear magnetic resonance, and (3) their in vitro or in vivo biological activities. These bioactive molecules constitute potential therapeutic or mechanistic tools that could be used to better understand the role of certain hormones in steroidogenesis.

## 1. Introduction

The use of a small molecule to block the activity of a key enzyme involved in the biosynthesis of certain hormones producing a beneficial or sometimes harmful effect, is a well-known therapeutic strategy in the biomedical field. Among the different families of hormones, the sex steroid hormones (glucocorticoids, mineralocorticoids, progestogens, androgens, and estrogens) are produced from cholesterol (Figure 1) obtained from exogenous (food) and endogenous (biosynthesis) sources [1,2,3]. The synthesis of these hormones is a multi-enzyme process called steroidogenesis and constitutes fertile ground for the identification of therapeutic targets, such as steroid sulfatase (STS), 17α-hydroxylase-C17,20-lyase (CYP17A1), the family of 17β-hydroxysteroid dehydrogenases (17β-HSDs), aldo-keto reductase (AKR)1C3, aromatase, and the 5α-reductase (5α-R) family. The development of inhibitors of aromatase and 5α-R, and more recently of CYP17A1, have given rise to drugs to treat certain hormone-dependent diseases [4,5,6,7,8,9], while the development of inhibitors of STS, AKR1C3, and some 17β-HSDs have not yet reached the drug stage, despite significant work in the last years [10,11,12,13,14,15,16,17,18,19,20]. Much less known, however, is the development of enzymatic activity enhancers to upwardly modulate hormone levels. Although few, activators constitute a new therapeutic approach that is complementary to that of inhibitors [21].

This account-type article aims to summarize our work which has made it possible to obtain, from a commercially available and affordable steroid, estrone (E1), inhibitors of the STS, 17β-HSD1, 17β-HSD2, 17β-HSD3, 17β-HSD5 (AKR1C3), and 17β-HSD12, as well as an activator of 17β-HSD12 (Figure 2). Having reached an advanced stage of development, these bioactive molecules constitute potential therapeutic or mechanistic tools that could be used to better understand the role of certain hormones in steroidogenesis. To facilitate their use by the scientific community, seven inhibitors (compounds **1**–**7**) and one activator (compound **8**) were then selected to discuss three important aspects of their development, namely, chemical synthesis, characterization, and identification of NMR markers, as well as the biological results generated by these bioactive steroid derivatives.

## 2. Chemical Synthesis of Inhibitors 1–7 and Activator 8

Inhibitor 1: The first part of the chemical synthesis of STS inhibitor **1** consisted of introducing a methoxy group at position C-2 of E1 (**9**) and generating the key intermediate **13** where the A-ring OH group was protected as a benzylic ether (Figure 1). The ortho-formylation of **9,** according to the method of Casiraghi et al. [22], which uses tin tetrachloride and paraformaldehyde, generated 2-formyl-E1 (**10**). After the protection of the OH group by benzyl bromide in the presence of cesium carbonate as a base to obtain the benzylic derivative **11**, a selective Bayer–Villiger-type oxidation made it possible to transform the formyl at C-2 into an aryl ester. This intermediate was hydrolyzed under acidic conditions in methanol. The resulting phenolic derivative **12** was then methylated with an excess of methyl iodide to obtain **13**. The last part of the synthesis began with an alkylation at position 17α. Despite the presence of a hindered ketone, such as for steroid **13**, the introduction of the benzyl group was achieved using a Grignard reaction [23]. Indeed, this reaction works very well for a benzylic derivative not having a β-hydrogen, unlike the alkyl Grignard reagents, which have a β-hydrogen [24]. Moreover, the presence of the axial methyl-18 in the alpha of the carbonyl favors an attack that takes place stereoselectively by the α-face of the steroid to obtain the 17α-benzyl derivative **14**. Under the conditions of catalytic hydrogenation, the debenzylation of the ether **14** made it possible to obtain the phenolic derivative **15**. Finally, since the inhibitory potential of aryl sulfamates for STS was abundantly documented [10,11,12,13,14,15,25,26], we then used sulfamoyl chloride in the presence of 2,6-di-*t*-butyl-4-methyl pyridine to obtain the STS inhibitor **1** [27].

Inhibitor 2: The chemical synthesis of 17β-HSD1 reversible inhibitor **2** begins with an aldol condensation of *m*-carboxamidobenzaldehyde and estrone (**9**) to obtain the α,β-unsaturated ketone **16** (Figure 2). The *m*-carboxamidobenzaldehyde building block is not commercially available, but it is easily obtained by acid hydrolysis from *m*-cyanobenzaldehyde [28]. Stereoselective reduction of the hindered ketone by the presence of a tertiary carbon and an axial methyl on the β-face of steroid **16** directs the hydride attack on the α-face to generate 17β-alcohol **17**. Given the steric hindrance of the β-face of compound **17**, the catalytic hydrogenation of the exo-alkene at C-16 takes place via the addition of hydrogen via the α-face, which leads to the formation of the 17β-HSD1 reversible inhibitor **2** [29].

Inhibitor 3: Several synthetic routes have been developed to reduce the number of chemical reactions (from 10 to 6 steps) necessary for the preparation of 17β-HSD1 irreversible inhibitor **3**, which has allowed to maximize the overall yield by limiting the purification work [30]. The reaction sequence optimized for the synthesis of **3** is shown in Figure 3. E1 (**9**) is first treated with trifluoromethanesulfonic anhydride to obtain E1-triflate (**18**). Then, a Suzuki-Miyaura coupling between **18** and potassium (2-benzyloxyethyl)trifluoroborate, prepared in one step using a published procedure [31], made it possible to introduce the 2-benzyloxyethyl chain in position 3 and to obtain **19**. From **19**, and following the sequence of three steps (aldol condensation, carbonyl reduction, and double-bond hydrogenation) previously reported for the synthesis of inhibitor **2** [29], the 16β-benzylcarboxamide side chain was introduced. The last two steps were, however, modified by (1) using NaBH_4_ at −40 °C rather than at room temperature to maximize the selective reduction of enone derivative **20** with the formation of 17β-OH derivative **21** and (2) changing palladium 10% on charcoal (Pd/C) for palladium hydroxide (20%) on charcoal (Pearlman’s catalyst) [32] to reduce the reaction time during the catalytic hydrogenation of **21** to **22**. In addition to providing a compound with the right C16β-stereochemistry, the palladium catalytic hydrogenation conditions had the advantage of removing the benzyl ether protecting group, thus generating diol **22**. Since the classic Appel reaction conditions (CBr_4_ and PPh_3_) provided an unselective bromination reaction of the primary alcohol of **22** over carboxamide and 17β-OH functionalities, we used the bromination reaction conditions developed by Chen et al. [33] for the deoxygenation of alcohols with tetrabutylammonium iodide and PPh_3_ in heated dibromoethane. Under these conditions, we obtained a satisfactory yield (83%) for the bromination of primary alcohol **22** to the 17β-HSD1 irreversible inhibitor **3** [34].

Inhibitor 4: The chemical synthesis of 17β-HSD2 inhibitor **4** begins with the protection of the phenol of E1 (**9**) with *t*-butyldimethylsilyl (TBDMS) chloride to form the silyl ether **23** (Figure 4). But-4-yn-1-ol is then protected in the form of a tetrahydropyrane (THP) derivative and then reacted with *n*-BuLi to generate the corresponding lithium acetylenide. The latter is then added to the carbonyl of TBDMS-E1 (**23**) at a very low temperature to promote the attack by the less hindered α-face and, thus, obtain alcohol **24**. Catalytic hydrogenation to reduce the triple bond and the subsequent hydrolysis of the THP group generated the diol **25**. Oxidation of the primary alcohol using Jones’ reagent yielded a carboxylic acid which did not cyclize to form the expected spiro-δ-lactone **26**. TBDMS group of **26** was cleaved by Bu_4_NF treatment to obtain the 17β-HSD2 inhibitor **4** [35].

Inhibitor 5: The chemical synthesis of 17β-HSD5 inhibitor **5** is similar to inhibitor **4,** except for the first and last steps (Figure 5). The E1 (**9**) hydroxy group was first esterified with trifluoromethanesulfonic anhydride and pyridine to the corresponding E1-triflate (**18**), which was reduced in the presence of triethylamine, formic acid, PPh_3,_ and palladium diacetate to obtain the deoxygenated compound **27**. The next four steps were previously described for the synthesis of **4** and consisted of (1) an alkylation with the appropriate lithium acetylenide, (2) a triple-bond hydrogenation, (3) a hydrolysis of THP protecting group, and (4) an oxidation to carboxylic acid that underwent an intramolecular cyclization with the tertiary alcohol. Using lithium diisopropylamide (LDA) as a base and methyl iodide in excess, the spiro-δ-lactone **30** was methylated in the α-position of the carbonyl. After the separation of the mono and dimethylated compounds by chromatography, compound **5** (diCH_3_) was isolated in moderate yield [36]; however, it was possible to increase this yield by methylation of the mono-methylated compound, obtaining 17β-HSD5 inhibitor **5**.

Inhibitor 6: 17β-HSD3 inhibitor **6** was synthesized in three steps, as reported in Figure 6. E1 (**9**) was first transformed into E1-triflate (**18**) with trifluoromethanesulfonic anhydride and pyridine. The carboxylic derivative **31** was generated by palladium-catalyzed hydrocarbonylation of aryl triflate **18** with Pd(OAc)_2_, KOAc, 1,1′-bis(diphenylphosphino)ferrocene (dppf) in DMSO under an atmosphere of carbon monoxide, as reported by Cacchi and Lupi [37]. On the other hand, amine **32** was obtained in one step by reacting 2-(trifluoromethyl)benzenesulfonyl chloride with an excess of *trans*-2,5-dimethylpiperazine. Finally, the formation of an amide bond between the carboxylic acid **31** and the amine **32** using hexafluorophosphate benzotriazole tetramethyl uronium (HBTU) as a coupling agent provided 17β-HSD3 inhibitor **6** [38].

Inhibitor 7: 17β-HSD12 inhibitor **7** was synthesized in four steps, as reported in Figure 7. After the transformation of E1 (**9**) into E1-triflate (**18**), followed by reductive deoxygenation to provide the deoxygenated compound **27**, a Samarium-Barbier reductive alkylation using 4-iodobenzyl bromide, samarium metal, and catalytic HgCl_2_ allowed to obtain the 17α-*p*-iodophenyl derivative **33**. Herein, the classic Grignard reaction was not appropriate to synthesize **33** because the formation of the needed magnesium species is not possible in the presence of the aromatic iodide. Since the palladium-catalyzed amidocarbonylation of aryl halides is a highly efficient, selective, and useful method for the direct synthesis of amide via coupling aryl halides with primary and secondary amines [39], the amidocarbonylation reaction of aryl iodide **33** and dimethylamine with high-density microwave was used to obtain compound **7**. The reaction carried out at 170 °C for 25 min using Mo(CO)_6_ as the CO source and palladacycle as the catalytic system provided a moderate yield of 17β-HSD7 inhibitor **7** [40]. 

Activator 8: 17β-HSD12 activator **8** was obtained as reported in Figure 8 by adapting a diversity-oriented synthesis (DOS) methodology reported for synthesizing various fused steroidal azacycles [41]. The 3-methoxymethyl-*O*-E1 (**34**), synthesized from E1 (**9**), was submitted to an aldol condensation with 3-formyl-benzonitrile to obtain the intermediate enone **35**. As previously reported for other steroidal-conjugated ketones, the carbonyl of **35** was stereoselectively reduced with sodium borohydride to obtain the 17β-OH derivative, which was treated with a solution of *m*-chloroperbenzoic acid in chloroform. After separating both epoxide derivatives by chromatography, the major α-epoxide **36** was heated under microwave at a high temperature with butylamine to afford the aminodiol intermediate **37**. Finally, the cyclization between the secondary amine and the OH group using triphosgene provided the carbamate E-ring derivative **38**, which after acid hydrolysis of the 3-methoxymethylether protecting group, provided 17β-HSD12 activator **8** [42].

## 3. NMR Characterization of Inhibitors 1–7 and Activator 8

Confirmation of the structure of compounds **1**–**8** was carried out using infrared (IR), mass (MS), and nuclear magnetic resonance (NMR) spectroscopies; however, the analysis of NMR data provided much more information. Indeed, although the IR data allowed the identification of the key functional groups and the exact high-resolution MS data confirmed the molecular formula of the desired steroid derivatives, NMR data made it possible to confirm the structural arrangement of all compounds. In addition, the presence of an E2 nucleus, whose NMR spectral data are very well known [43,44] (Table 1), facilitated the characterization work. Finally, in some cases, the use of two-dimensional (2D)-NMR experiments, such as COSY, NOESY, HSQC, and HMBC, proved crucial for identifying all the signals from which excellent markers were identified (Table 1 and Table 2). In this section, we will emphasize the addition of the chemical groups required for the development of the various inhibitors **1**–**7** and activator **8** by identifying their characteristic NMR (^1^H/^13^C) signals.

Inhibitor 1: For STS inhibitor **1**, the signals at 3.89 (^1^H) and 56.4 (^13^C) ppm confirm the presence of a methoxy group on the aromatic ring of the steroid A ring (Table 1). Moreover, the presence of only two aromatic CH in the form of two singlets indicates its positioning in C-2. In the NOESY and HSQC spectra, correlations with CH_2_-6 (2.82 ppm) make it possible to identify CH-4 (6.96/124.1 ppm) and, thereby, CH-1 (7.06/110.4 ppm). On the other hand, the protons of the sulfamate group (ArOSO_2_NH_2_) at 4.96 ppm, although present, are of little significance, given that they are labile and their intensity is variable. However, using known ^13^C NMR data from 2-methoxy-E2 shows that adding a sulfamate group at C-3 induces a shielding effect of ~5 ppm at C-3 and a deshielding effect of 6–9 ppm at C-2, C-4, and C-10 [27]; it was possible to identify the same effect for compound **1**. Finally, the presence of a benzyl group is confirmed in ^1^H NMR by the multiplet (5 CH) in the aromatic region (7.33 ppm) and in ^13^C NMR by the presence of four additional aromatic CHs (C-1″ to C-6″) and a CH_2_ at 42.3 ppm (C-1′). In ^1^H NMR, two benzylic protons (CH_2_-1′) are magnetically different and appear in the form of an AB system, i.e., 2 doublets (d) at 2.68 and 2.94 ppm. In ^13^C NMR, the replacement of the signal at 220 ppm characteristic of a carbonyl by another at 83.0 ppm (C-17) of a tertiary carbon carrying an alcohol is in line with the expected result of the Grignard reaction. Finally, the analysis of the HMBC spectrum makes it possible to observe key correlations between CH_3_-18 (s, 0.98 ppm) and C-17 (^3^*J*), as well as between CH_2_-1′ and C-17 (^2^*J*). Moreover, and in agreement with the addition of a Grignard reagent on the hindered carbonyl at steroid position C-17, the absence of a correlation between CH_3_-18 and CH_2_-1′ in the NOESY spectrum confirms the 17α-orientation of the benzyl group.

Inhibitor 2: For the reversible 17β-HSD1 inhibitor **2**, the NMR data slightly differ from those of E2, with the only significant difference being the presence of the *m*-carbamoylbenzyl group at position C-16β (Table 1). In ^1^H NMR, the addition of this group is supported by the presence of four new aromatic CHs (CH-2″, 4″, 5″ and 6″) and the signals of CH_2_-1′ appearing in a region that superimposes certain protons of the steroid nucleus. These signals, as well as those of labile CONH_2,_ are, therefore, not very characteristic. More characteristic is the signal at 3.81 ppm (CH-17) in the form of a doublet, which confirms the presence of a C-16 substituent. Among other things, in the NOESY spectrum, the absence of a correlation between CH_3_-18 (0.89 ppm) and CH-17 confirms the 17α-orientation of CH, while a correlation between CH-17α and CH-16 (identified in the COSY spectrum at 3.14 ppm) confirms the 16β-orientation of the *m*-carbamoylbenzyl group. The coupling constant for CH-17α (*J* = 9.6 Hz) is also consistent with the small angle between CH-17α and CH-16α. The analysis of the ^13^C NMR spectrum combined with those of the HMBC and HSQC spectra made it possible to easily identify CH_2_-1′ (38.9 ppm), CH-17α (83.0 ppm), and CH-16α (43.3 ppm), as well as the influence of a C-16 group on ring D carbons. In agreement with an NMR (^1^H and ^13^C) study for the 17-, 16-, and 15-allyl-E2 derivatives [45], signals at 3.81 (doublet) and 83.0 ppm are excellent markers for 17β-OH/16β-R stereochemistry. Furthermore, four CH and two C aromatic signals (129.1–144.3 ppm), as well as the amide signal at 272.7 ppm, showed characteristic correlations in (2D)-NMR spectra, confirming the presence of the *m*-carbamoylbenzyl group.

Inhibitor 3: For the irreversible 17β-HSD1 inhibitor **3**, the only difference with compound **2** is the replacement of the OH at C3 of steroid A-ring by a short 2-bromoethyl side chain. The spectral data are, therefore, identical for B, C, and D-rings, as well as for the *m*-carbamoylbenzyl group at C-16β (Table 1). In ^1^H and ^13^C NMR, the signals at 3.55/33.9 ppm and 3.06/40.1 ppm, whose HMBC, HSQC, and NOESY spectra analyses allowed the assignment of BrCH_2_-2‴ and CH_2_-1‴, respectively, are key markers. In ^1^H NMR, the replacement of OH at C-3 by the bromoethyl chain caused a deshielding effect on the three CH of A-ring (+0.52, 0.50, and 0.16 ppm for CH-4, CH-2, and CH-1, respectively). In ^13^C NMR, a strong shielding effect is observed on ipso C-3 (−18.2 ppm), while non-significant effects are observed on meta-positioned C-1 and C-5. For C-2 and C-4 located in ortho, the deshielding effect is significant (+14.5 and 15.2 ppm, respectively), while the effect is weaker (+9.6 ppm) for the C-10 in para.

Inhibitor 4: For the 17β-HSD2 inhibitor **4**, the ^13^C NMR data of A and B-rings (Table 2) are identical to those of E2 (Table 1); however, the introduction of a spiro-δ-lactone at position C-17 affects the chemical shifts of the D-ring. As expected, the most affected carbon is C-17 (92.4/+12.4 ppm), while C-13, C-16, and C-18 are less affected (−3.9, 3.5, and 2.9 ppm, respectively). Conversely, a deshielding effect was observable for C-12 (+5.1 ppm) and C-14 (−3.0 ppm). From the singlet easily attributable to CH_3_-18 (0.90 ppm) and thanks to the correlations observed in the HMBC and HSQC spectra, it was possible to confirm the chemical shifts of CH_2_-12 (*^3^J*), CH-14 (*^3^J*), C-17 (*^3^J*) and C-13 (*^2^J*). Four new signals were also present in the ^13^C NMR spectrum, namely the carbonyl characteristic of a lactone (171.1 ppm), as well as three methylene groups. Using the carbonyl signal (C-4″) and the correlations in the HMBC and COSY spectra, it was possible to assign these last signals as CH_2_-1″, 2″, and 3″.

Inhibitor 5: For the 17β-HSD5 inhibitor **5**, there are two main structural differences compared to inhibitor **4** discussed above. The first, in the right portion of the molecule, is due to the presence of two methyl groups on the C-3′ in alpha to the carbonyl of the spiro-δ-lactone. These two CH_3_ appear as a singlet at 1.28 ppm in ^1^H NMR and two signals at 27.5 and 27.8 ppm in ^13^C NMR (Table 2). Their presence causes a deshielding effect at C-4″ (177.8/+6.7 ppm), C-3″ (37.8/+8.8 ppm), and C-2″ (34.8/+19.5 ppm). The second difference, in the left portion of the molecule, is due to the replacement of the OH at C3 by an H to obtain a deoxygenated steroid A-ring. This modification is easily confirmed by the presence of a new aromatic CH whose chemical shifts (7.13 and 125.2 ppm) in ^1^H and ^13^C NMR are identical to those of CH-1 and CH-2.

Inhibitor 6: For the 17β-HSD3 inhibitor **6**, the basic nucleus is E1 (17-C=O), unlike that of E2 (17β-OH) used for the other inhibitors and the activator. The comparison of ^13^C NMR data (Table 2) for C and D-rings is, therefore, in agreement with those reported in the literature for E1 [43]. A functionalized side chain linked at position C-3 of the steroid by an amide bond influences the chemical shifts of the A-ring signals, mainly in ^13^C NMR. Thus, deshielding effects are observed on the carbons in ortho (C-2/+13.3 ppm and C-4/+12.0 ppm) and para (C-10/+6.0 ppm), as well as a shielding effect on the carbon carrying the chain (C-3/−13.3 ppm) compared to the data of C-1 to C-5 and C-10 of E2 (Table 1). The presence of the *o*-trifluoromethylphenylsulfonamide-*trans*-dimethylpiperazinoamide chain is confirmed by the appearance of several characteristic but complex signals, except for the carbonyl (C-1′) of the amide at 171.5 ppm. Thus, due to the presence of amide and sulfonamide bonds, as well as possible conformations, the substituted piperazine ring gives rise to several signals between 40.5 and 50.0 ppm (CH_2_ and CH) and 15.8 and 14.4/14.6 ppm (CH_3_) in ^13^C NMR, as well as broad signals between 3.30 and 4.90 ppm (CH_2_, CH) and 1.06 and 1.19 ppm (CH_3_) in ^1^H NMR. The phenyl ring of the side chain gives rise to four aromatic CHs (7.70–8.18 ppm) in ^1^H NMR in addition to their corresponding CH and C signals (C-1‴ to CH-6‴) in ^13^C NMR. The presence of three fluorine atoms (CF_3_) causes the characteristic splitting of carbons carrying or neighboring the fluorine atom in the form of a quadruplet (q). Among other things, CF_3,_ therefore, appears at 122.5 (q, ^2^*J_CF_* = 274 Hz), C-2‴ at 127.2 (q, ^3^*J_CF_* = 32 Hz), and C-3‴ at 128.6 (q, ^4^*J_CF_* = 6.4 Hz).

Inhibitor 7: For the 17β-HSD12 inhibitor **7**, the characteristic signals of its 3-desoxy E2-nucleus are the same as those of inhibitor **5**. In ^1^H NMR, the addition of the *p*-dimethylamidobenzyl group is confirmed by two doublets (AB system) of 17α-CH_2_ (C-1′), two singlets at 3.05/3.13 ppm (N(CH_3_)_2_), and a doublet at 7.39 ppm of four aromatic CHs. In ^13^C NMR, the signals at 35.4 and 39.7 ppm are associated with the N-dimethyl amide group but are very weak. The signal at 83.2 ppm is attributed to C-17, identical to the one observed for inhibitor **1**, which has a 17α-benzyl group. Moreover, the lack of correlations between CH_3_-18 and CH_2_-1’ in the NOESY spectrum supports the 17α-orientation of the benzyl substituent.

Activator 8: For the 17β-HSD12 activator **8**, the signals associated with the A, B, and C-rings (Table 2) are the same as those of E2 and inhibitors **2** and **4** (Table 1 and Table 2). Although the expected signals are found for the *m*-carbamoylphenyl group (127.5–139.6 and 167.1/168.7 ppm), as well as the N-butyl chain (49.2, 30.1, 20.5, and 14.1 ppm), the substitutions at positions C-16 and C-17 greatly affect the chemical shifts in D-ring. Three markers have been identified: C-16, which is strongly deshielded at 82.8 ppm due to the presence of tertiary OH, as well as CH_2_-15 and C-17, deshielded by an alpha substitution at 38.6 and 97.9 ppm, respectively. Two other interesting markers are associated with the new E-ring: carbamate carbonyl (OCON) at 156.4 ppm and CH-1′ at 72.0 ppm. In ^1^H NMR, singlets at 4.12 ppm (CH-17) and 4.77 ppm (CH-1′) are also characteristic and have been used for the assignment of stereochemistry at positions C-17 and C-16.

## 4. Biological Activity of Inhibitors 1–7 and Activator 8

Inhibitor 1: Steroid sulfatase (STS) is a widely distributed microsomal enzyme in human tissues that catalyzes the hydrolysis (desulfation) of sulfated 3-hydroxy steroids, the inactive form of steroid hormone, or steroid precursor, to the corresponding free active 3-hydroxy steroids [46,47,48]. STS plays a crucial role in the formation of steroid hormones, and its inhibition constitutes a therapeutic approach envisaged to better control blood and tissue hormone levels, thus opening the door to potential new therapies [10,11,12,13,14,15].

The STS irreversible inhibitor **1** stems from work showing that the presence of a hydrophobic group, such as the benzyl group, added to position C-17α of E2 induced a reversible inhibition of STS [23,49]. Since it was known from the work of Reed and collaborators that the addition of a sulfamate group (NH_2_SO_3_) at C-3 of E1 generated EMATE, a potent reversible inhibitor of STS [50], 17α-benzyl-E2 was sulfamoylated to obtain an irreversible STS inhibitor, which is very effective in inhibiting the formation of E1 and DHEA from E1S and DHEAS [51]. In addition, using the 2-methoxy-E2 ring then made it possible to block the undesirable residual estrogenic activity of this family of STS inhibitors and obtain the STS irreversible inhibitor **1** [27]. In fact, **1** strongly inhibited (IC_50_ = 0.024 nM) the transformation of E1S into E1 when tested in homogenized HEK-293 cells overexpressing STS (Table 3). It did not show estrogenic and antiestrogenic effects in the uterus (an ER+ tissue) of ovariectomized mice, but when given subcutaneously (*sc*) or orally (*po*), it reversed the estrogenic effect caused by E1S [27]. Similarly, it showed no androgenic effect in the ventral prostate and seminal vesicles (two AR+ tissues) of castrated rats and reversed the androgenic effect caused by DHEAS [52]. In addition, inhibitor **1** completely blocked the growth of E2S-stimulated breast cancer tumors in mice, thus demonstrating its potential as an anticancer agent [53].

Inhibitors 2 and 3: 17β-Hydroxysteroid dehydrogenase type 1 (17β-HSD1) is a cytosolic enzyme involved in the last step of the biosynthesis of E2, the most potent estrogen, from E1 [54,55,56]. Using DHEA as substrate, this enzyme is also involved in the formation of 5-diol, a weak estrogen that becomes more important at menopause, when the ovaries no longer produce E2. Inhibitor **2** (CC-156) effectively blocks the conversion of E1 to E2 by purified human 17β-HSD1 (K_i_ = 0.9 nM), as well as by the 17β-HSD1 activity present in human T47D cancer cells (IC_50_ = 27–44 nM) [29], and this by competing with E1 in the catalytic site, as shown by the 3D analysis of the 17β-HSD1/CC-156 complex [57] (Table 3). Despite its ability to reverse the proliferation of ER+ T47D cells induced by E1, inhibitor **2** also exerts an estrogenic effect when tested alone in this cell model, as well as in mice [58]. Since this residual estrogenic activity of reversible competitive inhibitor **2** is not compatible with its intended therapeutic use, a structure-activity relationship (SAR) study subsequently made it possible to identify inhibitor **3** (PBRM) [59]. Here, the replacement of the 3-OH of **2** (CC-156) by a short 2-bromoethyl chain made it possible to obtain the irreversible competitive inhibitor **3**, as shown beyond all doubt by the 3D analysis of the 17β-HSD1/PBRM complex (Figure 3) [60]. Indeed, we observed the formation of a covalent bond between a nitrogen atom of histidine-221 of human 17β-HSD1 and the methylene group (CH_2_) of the short side chain (displacement of Br). It is, therefore, an irreversible inhibitor (K_i_ = 368 nM and K_inact_ = 0.087 min^−1^) when tested in pure human enzyme but selective for 17β-HSD1 (mild alkylating agent) [61]. Inhibitor **3** is less potent than inhibitor **2** in T47D cells (IC_50_ = 68 and 27 nM, respectively), but unlike its reversible analog **2,** it did not show estrogenic activity in mice [58]. Thanks to these characteristics, inhibitor **3** was able to effectively block the growth of tumors (xenografts of T47D human cancer cells in nude mice) induced by two precursors of E2, namely E1 and DHEA [62], and has also shown its efficacy in a monkey model of endometriosis [63].

Inhibitor 4: 17β-Hydroxysteroid dehydrogenase type 2 (17β-HSD2) is a membranar enzyme involved in the oxidation of key hydroxysteroids, such as E2, 5-diol, T, DHT, and 20α-DHP, to less active corresponding ketosteroids [64]. Blocking 17β-HSD2 activity would keep the levels of certain steroid hormones high enough to benefit from desirable estrogenic or androgenic effects to treat some hormonal deficiencies, as in the case of osteoporosis. The predominant activity of 17β-HSD2 is mainly the oxidation in whole cells; however, both oxidative and reductive activities were observed in homogenized transfected HEK-293 cells according to the cofactor used, NAD(P)+ or NADP(H), respectively. From a SAR study using the reductive activity of 17β-HSD2, the spiro-δ-lactone-E2 compound **4** was identified as the most potent inhibitor for the transformation of 4-dione to T (IC_50_ = 6 nM; K_i_ = 29 nM) in a non-competitive and reversible manner (Table 3) [35]. When tested against the oxidase activity of 17β-HSD2, compound **4** was also able to inhibit the transformations of both T to 4-dione and E2 into E1 (65% at 1 μM).

Inhibitor 5: 17β-Hydroxysteroid dehydrogenase type 5 (17β-HSD5), or aldo-keto reductase (AKR) 1C3, catalyzes the conversion of 4-dione into T, DHEA into 5-diol, and E1 into E2; consequently, its inhibition could be a strategy to lower the level of androgen T and estrogens E2 and 5-diol [19,20,65,66,67]. Inhibitor **5** was obtained by modifying 17β-HSD2 inhibitor **4**. To make the latter selective for 17β-HSD5, two modifications were necessary: adding two methyl groups in alpha to the carbonyl of the spiro-δ-lactone and using a deoxy-E2 ring. Compound **5** thus inhibited the reduction of 4-dione to T (IC_50_ = 2.9 nM) by transfected 17β-HSD5 in whole HEK-293 cells, but without inhibiting at 0.1 and 1 μM the oxidative and reductive activities of 17β-HSD2 present in homogenized transfected HEK-293 cells (Table 3) [36].

Inhibitor 6: 17β-Hydroxysteroid dehydrogenase type 3 (17β-HSD3), or testicular 17β-HSD, is expressed in the microsomal fraction of testes and is involved mainly in the oxidation of the androgenic hormone T to inactive 4-dione [67,68,69,70]. Compound **6** inhibited the transformation of 4-dione into T in both whole LNCaP cells overexpressing 17β-HSD3 (IC_50_ = 0.10 μM), as well as in a preparation of homogenized rat testes (IC_50_ = 0.11 μM) (Table 3). This inhibitor also reduced the proliferation of androgen-dependent (AR+) prostate cancer LAPC-4 cells but not the proliferation of (AR-) prostate cancer PC-3 cells, suggesting an effect mediated via the reduction of the production of androgens [38]. When injected *sc* in mice, inhibitor **6** was found to be more abundant (4 times) in plasma than the previous lead inhibitor RM-532-105, built around a C19-steroid backbone [71,72], thus suggesting a better metabolic stability of the E1 (C18-steroid)-type 17β-HSD3 inhibitor **6**.

Inhibitor 7: 17β-Hydroxysteroid dehydrogenase type 12 (17β-HSD12) was initially reported as 3-ketoacyl-CoA reductase for its essential role in fatty acid elongation cascade [73] but was later found to be able to reduce E1 into E2 [74]. Although its physiological role remains a matter of debate, inhibiting 17β-HSD12 could be a potential strategy to reduce the E2 levels where the enzyme is present. Inhibitor **7** originated from a SAR study showing that a side chain in position C-17α and the absence of an OH group at position C-3 of E2 are important for the inhibition of 17β-HSD12 activity [40]. Thus, at 1 and 10 μM, respectively, the dimethylamidobenzyl derivative **7** inhibited 71 and 82% of the transformation of E1 into E2 by 17β-HSD12 overexpressed in whole HEK-293 cells (Table 3) [75]. Having shown selectivity for type 12 over types 1, 2, 5, and 7 17β-HSDs, inhibitor **7** was used in a study to investigate the role of certain 17β-HSDs in the proliferation of breast cancer cells [76]. 

Activator 8: Medicinal chemists designed numerous enzyme inhibitors; however, the discovery of small molecule activators that interact with an enzyme to enhance its activity is a rather rare event. In fact, to date, only 12 enzymes have well-known synthetic activators [21]. By testing a series of steroid derivatives as 17β-HSD12 inhibitors, E2 derivatives with an [1,3]oxazinan-2-one E-ring were found to increase the E1 to E2 conversion basal activity when tested in intact and homogenized stably transfected HEK-293 cells. Thus, for compound **8** tested in transfected HEK-293 cells, a dose-dependent stimulation of E1 to E2 conversion was observed in homogenized cells (microsomal fraction) (EC_50_ = 7.5 μM), while a stimulation of 281% was measured at 20 μM in intact cells (Table 3). The exact mechanism of action of activator **8** is not yet known, but results using a microsomal fraction suggest an allosteric binding of **8** to 17β-HSD12. Stimulation of E1 to E2 conversion was also observed in T47D breast cancer cells which are known to highly express 17β-HSD12 [42].

**Table 3 molecules-28-03499-t003:** Summary of biological activities for compounds **1**–**8**.

Cpd	Type	Enzyme	IC_50_ (K_i_) in nM	EC_50_ in nM	Other ^1^	Ref
1	Inh	STS	0.024 ^2^	--	Irreversible Inh; Not estrogenic and not androgenic in mice and rats; Blocks E1S, E2S, and DHEAS transformations in cells and mouse/rat tissues	[27,52,53]
2	Inh	17β-HSD1	27–44 ^3^ (0.9)	--	Reversible Inh; Weakly estrogenic in cells and mice	[29,57]
3	Inh	17β-HSD1	68–83 ^3^ (368)	--	Irreversible Inh (K_inact_ = 0.087 min^−1^); Not estrogenic in cells and mice; Blocks E1- and DHEA-tumor growth in mice; Orally active	[58,59,60,61,62,63]
4	Inh	17β-HSD2	63 ^4^ (29)	--	Reversible Inh; Potentially estrogenic; NTAM	[35]
5	Inh	17β-HSD5 (AKR1C3)	2.9 ^5^	--	Not androgenic and weakly estrogenic in cells; NTAM	[36]
6	Inh	17β-HSD3	100–110 ^6^	--	Reduces androgen (AR^+^)-dependent cells; NTAM	[38]
7	Inh	17β-HSD12	560 ^7^	--	NTAM	[75]
8	Act	17β-HSD12	--	7500 ^8^	NTAM	[42]

^1^ Inh—Inhibitor; Act—Activator; NTAM—Not tested in an animal model; ^2^ Inhibition of E1S to E1 in homogenized transfected HEK-293[STS] cells. ^3^ Inhibition of E1 to E2 in whole breast cancer T47D cells. ^4^ Inhibition of 4-dione to T in homogenized HEK-293[17β-HSD2] cells; 65% of inhibition at 1 µM for T to 4-dione transformation. ^5^ Inhibition of 4-dione to T in whole HEK-293[17β-HSD5] cells. ^6^ Inhibition of 4-dione to T in both whole LNCaP[17β-HSD3] cells and homogenized rat testes. ^7^ Inhibition of E1 to E2 in whole HEK-293[17β-HSD12] cells. ^8^ Stimulation of E1 to E2 transformation in homogenized (microsomal fraction) of HEK-293[17β-HSD12] cells.

## 5. Conclusions

From the C18-steroid estrone (E1) as a starting material and using a short sequence of reactions, the introduction of a functionalized side chain at key positions, such as positions C-3, C-16, C-17, or both C-16 and C-17, made it possible to generate seven inhibitors (compounds **1**–**7**) and one activator (compound **8**) of steroidogenic enzymes [STS, 17β-HSDs 1, 2, 3, and 12, and AKR1C3 (17β-HSD5)], all involved in the biosynthesis of estrogens and androgens. Irreversible STS inhibitor **1** combines three important elements, namely, a hydrophobic group at C-17α, a sulfamate group at C-3, and a methoxy group at C-2 of estradiol (E2), resulting in a very potent and non-estrogenic inhibitor. The 17β-HSD1 inhibitors were obtained by adding an *m*-carbamoylbenzyl chain at C-16β of E2, while a judicious modification of the C-3 position (from OH to BrCH_2_CH_2_) made it possible to obtain inhibitor **2** (reversible) and inhibitor **3** (irreversible), the latter of which proved effective when tested in cancer and endometriosis animal models. 17β-HSD2 and 17β-HSD5 (AKR1C3) inhibitors share the common element of a spiro-δ-lactone ring at C-17 of E2, but the addition of two methyl groups at C-3′ and the absence of the OH at C-3 make inhibitor **5** selective for 17β-HSD5, unlike inhibitor **4**. The introduction at position C-3 of E1 of an arylsulfonamidodimethylpiperazine chain optimized during SAR studies led to 17β-HSD3 inhibitor **6**. For 17β-HSD12, a first family of inhibitors represented by compound **7** was obtained by adding a *p*-amido-benzyl chain in position C-17α of desoxy-E2, while diversification of the positions C-16 and C-17 made it possible to obtain compound **8**, a first activator of an enzyme involved in steroidogenesis.

In summary, the bioactive compounds **1**–**8**, for which chemical synthesis, NMR characterization, and biological activities have been described, constitute potential therapeutic or mechanistic tools that could be used to better understand the role of certain hormones in steroidogenesis and potentially contribute to the development of new drugs.

## Data Availability

Not applicable.

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
