# Peer review of "Description of Chemical Synthesis, Nuclear Magnetic Resonance Characterization and Biological Activity of Estrane-Based Inhibitors/Activators of Steroidogenesis"

_molecules, 2023, doi:10.3390/molecules28083499_

Round 1

Reviewer 1 Report

The article is well written, describes solid research, and reads like a very good original article – and as such would have represented an excellent experimental work.  However, this is essentially an account of the Author’s works.  Thirty-one references (i.e., more than 40%) are to these works.  This is not a review, because there is no clear reason for the selection of 8 compounds described in the paper, and their one-by-one description lacks proper context.  

In a comprehensive review, one would expect at least to learn how the compounds differentiate from the plethora of other compounds.  In a tutorial review, for example, a basic understanding of their synthesis and spectra would be best if the description was not fragmented into separate sections for each compound.

If a publication of an account-type paper (line 1 change Review to Account) is acceptable, then the title of the manuscript should be changed accordingly.

Minor comments:

AKR should be explained earlier in the text

The reviewer finds the use of plural “we” and “our” odd. (e.g., line 58)

m-cyanobenzaldehyde” in line 95 refers to carboxamide not nitrile.

Use proper labeling of coupling constants, i.e.,  4J or 4JC-F but not JCCCF

The first paragraph of conclusions does not convey conclusions from the main article body.

Author Response

Please see my Response to Reviewers' comments provided as a pdf file. Thank you.

Reviewer 2 Report

Donald described the review "Estrane-based inhibitors/activators of steroidogenesis for treating estrogen/androgen-dependent diseases: Chemical synthesis, nuclear magnetic resonance characterization, and biological activity" well written. The author explained the chemical synthesis, NMR characterization, and biological activities of steroid derivatives. Here the author compared the NMR data and explained the biological activity of steroid derivatives. So this review might use to design the new analogs. This review can accept after minor revision.

In Scheme 1, the authors should mention the reaction temperature from the conversion of 11 to 12. 

If the author provides a table with biological activity data, it would be more useful for readers.

Author Response

Please see my Response to Reviewers' comments. Thank you.

Reviewer 3 Report

1. The title of the review not justifying the work reviewed and it is too long.  The author should consider revising the title appropriately.

2. Page 1, line 14 what are those diseases?

3. On what basics the set of inhibitors were selected?

4. The motivation behind this review should be written clearly which is not quite coming in this version like the importance of this review; the author should state precisely.

Author Response

(The authors gave the same response as above.)

Round 2

Reviewer 1 Report

Neither the title nor the abstract reflect the account type of this paper.  Please change the title to reflect the actual type of the paper as suggested in the initial review.  Consider modifying the abstract as well. 

Very minor:

In Figure 1, the Author uses italic numbers, and the caption uses bold italic numbers for enzymes.  Later, similar bold numbers are used for steroids.  Consider changing the labeling in this Figure to bold letters. 

Please provide links to Ph.D. theses if available. I suggest adding a web address followed by an 'accessed on date' statement. 

Author Response

- In the title, "Description of" was added at the beginning of the title, which had already been modified and shortened during the first revision. The new title is: “Description of chemical synthesis, nuclear magnetic resonance characterization and biological activity of estrane-based inhibitors/activators of steroidogenesis”.

- The abstract (middle) has been modified by changing “This review article” by "This account-type article". We also reported “account-type article at the end of introduction (last paragraph). In addition, the abstract was satisfactory for Reviewers 2 and 3.

- In Figure 1, the italic numbers have now been reported in italic and bold characters as requested.

- For reference 75 (PhD Thesis), I followed the Instructions for Authors and added a web address.

I hope these new changes will satisfy Reviewer 1. Thank you.

Reviewer 3 Report

Satisfied with this version.

Author Response

The reviewer mentionned "Satisfied with this version".Thank you.